# Application of 3D Digital Image Correlation for Development and Validation of FEM Model of Self-Supporting Arch Structures

**Krzysztof Malowany** [1,*], **Artur Piekarczuk** [2] , **Marcin Malesa** [1], **Małgorzata Kujawińska** [1] and **Przemysław Więch** [2]

[1] Warsaw University of Technology, Institute of Micromechanics and Photonics, 8 Św. A. Boboli St., 02-525 Warsaw, Poland; m.malesa@mchtr.pw.edu.pl (M.M.); m.kujawinska@mchtr.pw.edu.pl (M.K.)

[2] Instytut Techniki Budowlanej, 1 Filtrowa St., 00-611 Warsaw, Poland; a.piekarczuk@itb.pl (A.P.); p.wiech@itb.pl (P.W.)

[*] Correspondence: k.malowany@mchtr.pw.edu.pl

**Abstract:** Many building structures, due to a complex geometry and non-linear material properties, are cumbersome to analyze with finite element method (FEM). A good example is a self-supporting arch-shaped steel sheets. Considering the uncommon geometry and material profile of an arch (due to plastic deformations, cross section of a trough, a goffer pattern), the local loss of stability can occur in unexpected regions. Therefore, the hybrid experimental-numerical methodology of analysis and optimization of arch structures have been proposed. The methodology is based on three steps of development and validation of a FEM with utilization of a digital image correlation (DIC) method. The experiments are performed by means of 3D DIC systems adopted sequentially for each measurement step conditions from small size sections, through few segment constructions up to full scale in situ objects.

**Keywords:** experimental-numerical method; digital image correlation; finite element method; static analysis; arch structures

## 1. Introduction

The development of large-scale and complex engineering structures creates new challenges for designers and constructors, who need to meet the demands of increasing safety, extended component lifetime and simultaneously reduced investment and operation costs. To fulfill these requirements new materials (e.g., composites) and assembly technologies are being developed. An example of this type of construction is a self-supporting arch structure. Such structures had been initially built as temporary buildings used for military purposes. Adaptations of this technology for civil purposes, required extension of the designed lifetime and consideration of different environmental conditions, and therefore design problems, especially in terms of stability and load transfers occurred [1–5]. In order to ensure safety and proper operational parameters during their lifetime, hybrid experimental-numerical methods are being used during design and exploitation stages [6]. The common practice in experimental mechanics is to validate the numerical model using point wise sensors (e.g., strain gauges). They are attached to a tested structure in places in which the highest stress concentrations are expected (based on the analysis of a numerical model). This is a simple and low-cost approach, but a problem can occur if an inaccurate numerical model does not indicate all the places in which stress concentrations occur. This, in turn, may cause errors in the process of validation of the numerical model. Therefore, an advanced evaluation of numerical model is more often supported with

the data obtained by means of full-field optical measurement methods, which determine displacements and deformations in critical areas of the objects [7,8]. In the case of investigations of large engineering structures, the most commonly used techniques are terrestrial laser scanning [9–11] and the 3D digital image correlation (3D DIC) method [1,12–17]. Terrestrial laser scanning systems enable 3D measurements of shape of object. By comparison of the acquired shapes in different load conditions, deformation of an object under the load can be calculated [10,11]. The method is simple to use, however, due to time required for measurement (from few second up to few minutes), its utilization is limited to static measurements. 3D DIC measurement systems combine digital image correlation and triangulation methods. 3D DIC provides directly 3D displacement vector **d** distribution (displacement maps (u,v,w) in x, y and z directions respectively) in a measured field of view [18]. Utilization of the 3D DIC method requires modification of a measured surface (applying a paint coat that provides a random texture) therefore it is more difficult to use compared to terrestrial laser scanning systems. Nonetheless the 3D DIC system enables measurements of an investigated object under varying in time conditions, and therefore this method have been used in the presented application. Numerical modeling of a self-supporting arch structure is cumbersome, considering the uncommon geometry and material characteristics (due to plastic deformations, the cross section of a trough, goffer pattern). Thus, the development and validation of a FEM model of a full-sized construction made of self-supporting arch sheets described in Section 2, was divided into three steps starting from small sized sections (Section 4), through few segment constructions (Section 5), to full scale in situ objects (Section 6). Each step required a tailored approach to the measurements with 3D DIC due to different: accuracies, sizes of fields of view, forms of outcome data, environmental conditions. Some aspects of this work had been presented in our previous papers. In papers [2,3] the investigation of 1 m long section of arch-shaped steel sheets was presented, the goals of this work was to simulate local loss of stability and to determine a geometric model of the surface shape for FEM analysis. In papers [1,13] the investigation of segments of arch-shaped steel sheets in laboratory conditions were presented; the goals of this work were to investigate the global stability and to determine the mechanical behavior of supports. The measurements presented in paper [13] were performed with utilization of multi-camera DIC system with overlapping areas of fields of view of cameras. In paper [16] the performed measurements of the full-scale construction made of arch-shaped self-supporting metal plates was presented, and in order to enable these measurements the multi-camera DIC system with distributed field of view (FOV) was developed. In this paper we summarize the development of 3D DIC systems and present the combined three-steps hybrid experimental-numerical methodology of analysis and optimization of arch structures.

## 2. Characteristic of the Investigated Object and General Procedure Supporting Development and Validation of a Finite Element Model

### 2.1. Specification of Arch-Shaped Steel Sheets Used as a Self-Supporting Arch Structures

A self-supporting arch-shaped covering of steel sheet sections is used in civil engineering [19]. Typical radii of the assembled arch coverings varies in the range of 6 m to 30 m (Figure 1a). Due to the simple design, quick installation and relatively low implementation costs, this type of covering gained significant popularity. Arch-shaped steel sheets are cold formed in two stages. At first, the flat metal plates (stored in a roller) are bended in order to receive plates with trapezoidal cross section, then the plates are goffered (local plastic deformation) in order to receive arches (Figure 1b). A full scale object is obtained from a number of single arches connected to each other with double lock standing seams. Considering the trapezoidal cross section, the differences in radius of the arch cause waiving in the surface between shelves.

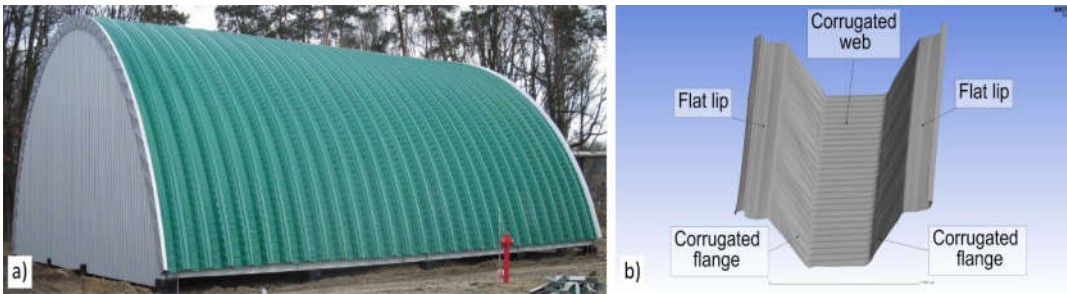

**Figure 1.** (**a**) Outside image of the measured hall as an example of typical self-supporting arch-shaped covering, (**b**) section surfaces.

### 2.2. Three-Step Development and Validation of a Numerical Model

Considering the uncommon geometry and material characteristic of an arch, development of a FEM model of such a structure required a specific approach and its validation at different stages of advancement. The development of a FEM model of the full-sized construction made of the self-supporting arch has been heavily supported by the experiments performed by means of 3D DIC within three steps. At first, the tests have been carried out on single sections of the arch in order to determine which geometric model of the surface shape (planar, corrugation or corrugation and wavy model) should be applied in the further steps for FEM analysis. The considered sections were 1 m long and 0.7 m wide. In the next step, a structure composed of four individual segments with the geometry selected at step 1 has been used in laboratory tests with controlled loading conditions in order to investigate the global stability and to determine the mechanical behavior of supports, in particular to define the rotational stiffness of supports used in a FEM model of arches. The measured object was of 12 m span and 2.8 m wide. The final step was based on the tests of a full-scale object: 8 m high covering the area of 18 × 18 m. The outcomes of this test is the validation of a complete FEM model in which the knowledge gained in steps 1 and 2 had been implemented. It has been proven that the model accurately simulates construction deformations under the environmental loads. The model can be scaled to larger constructions. The general procedure supporting the development and validation of a FEM model has been summarized and presented in Figure 2. This procedure is described in detail in Sections 5 and 6.

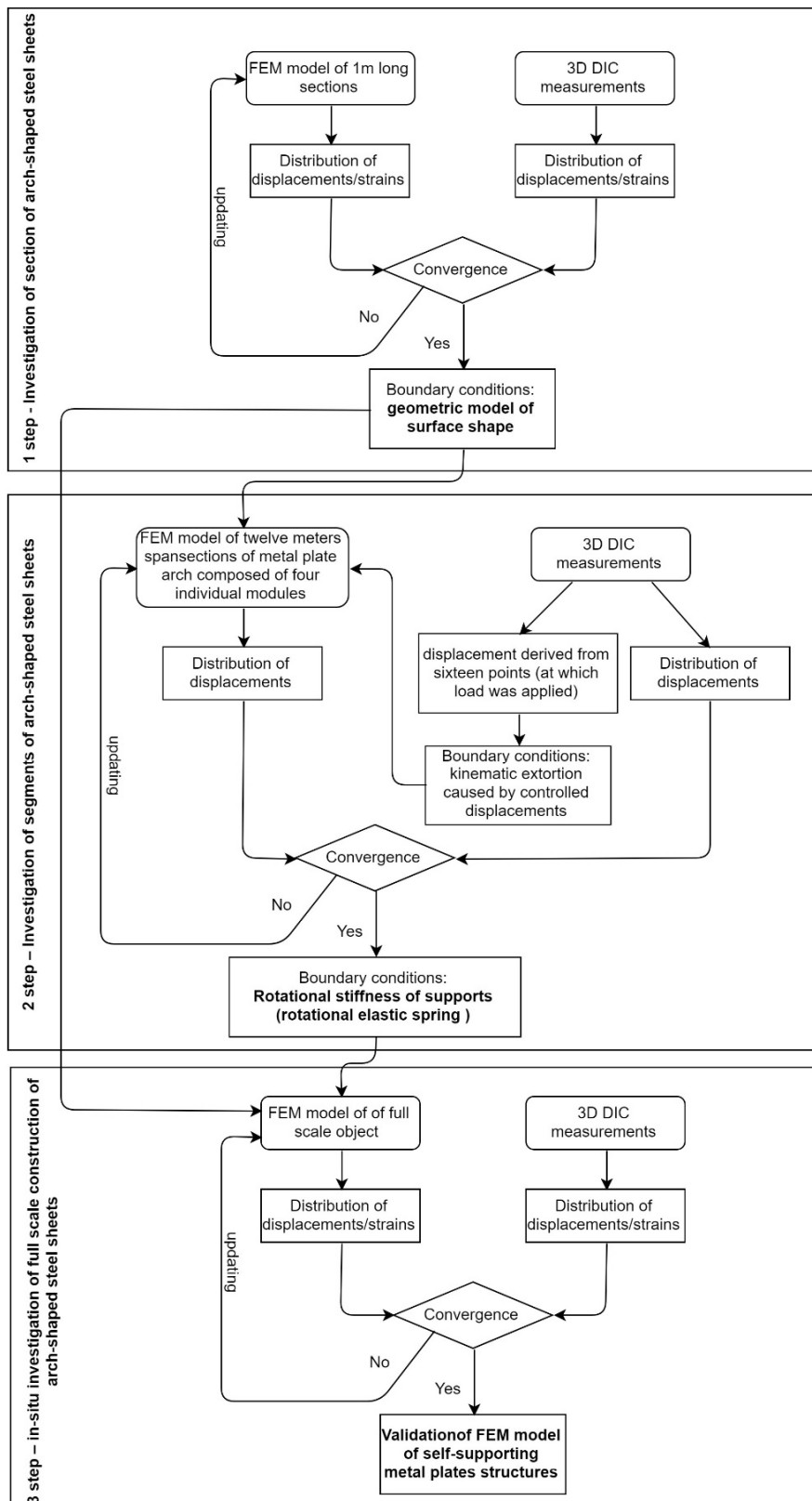

**Figure 2.** Flowchart of the 3-steps updating and validation of numerical model of self-supporting arch structures with utilization of 3D digital image correlation method.

## 3. Digital Image Correlation Method

At each stage of the procedure described in Section 2 full-field measurements of displacement vector **d** (u,v,w) of the investigated structures are required. The method which is best fitted to the measurement requirements is digital image correlation [18]. DIC is based on acquisition of a set of images of a tested object which is subjected to load. The surface of an object under investigation has to be covered with a random texture. The 2D DIC version uses a single camera. One of the acquired images is selected as a reference for the others. The reference image is divided into small regions (or subsets), a position of each subset is tracked in the remaining images, using the maximum zero-mean normalized sum of squared difference function as the criterion (or any other correlation metric). The image can be divided into hundreds or thousands of subsets, thus 2D DIC provides in-plane displacement maps over the selected area of interest (AOI). The 3D DIC is a technique that combines the 2D DIC with stereovision by using two cameras for observation of the same AOI. 3D DIC provides: the 3D shape of a surface, in-plane and out-of-plane displacement maps. The in-plane strain maps are calculated by differentiation of the in-plane displacement maps. According to [20], the minimum displacement measurement error can be less than 0.001 pixels, however, it must be noted that in real applications the accuracy of measurements strongly depends on factors such as image noise and stability of experimental conditions. Moreover, the out-of-plane measurement error is larger than the in-plane one, and strongly depends on a stereo angle between the cameras as set [20,21]. The accuracy of displacement measurements is scalable with the resolution of the cameras (larger camera's resolution and smaller FOV indicate higher displacement measurement accuracy). The displacements in the "x" direction are given as "u" in [pixels] and after scaling are expressed as "U" in [mm]; similarly, displacements in the "y" and "z" direction are given as "v [pixels] ", "V[mm] ", and "w[pixels] ", "W[mm] ", respectively.

## 4. Investigation of Section of Arch-Shaped Steel Sheets

At this stage, the test bench was designed in order to simulate local loss of stability in 1 m long sections [2,3]. The examined sample was fastened between two horizontal rigid plates, with defined degrees of freedom. The force was eccentrically applied in the direction of the axis that passes through the center of gravity of the cross sections. With this arrangement, the compression force on 1 m long interval of the arch (having a radius of 18 m) was mapped.

### 4.1. 3D Digital Image Correlation Setup

The 3D DIC system used in this measurements comprises two AVT Pike F-1600 (4872 × 3248 pixels) monochromatic cameras equipped with 28 mm lenses, set on an angle of 30°. The setup was mounted on an aluminum frame to enable easy geometric modifications. The surface of the examined specimen was illuminated with two 200 W light-emitting diode (LED) lamps (13,000 lumen) equipped with a light diffuser ("soft box"), in order to eliminate shadows on the surface. The FOV of the system was 1.5 m × 1 m, covering the entire area of the sample (Figure 3).

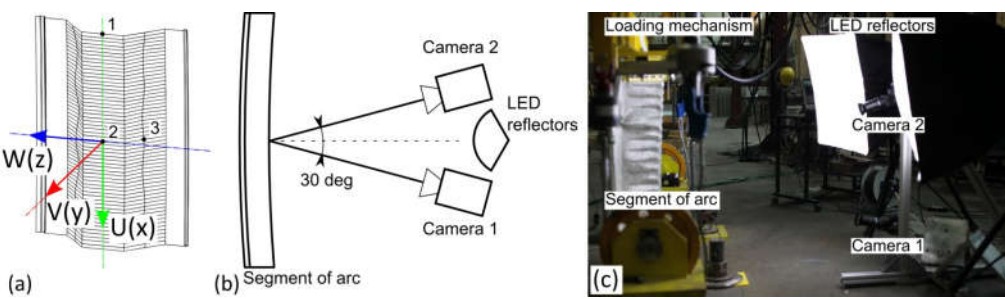

**Figure 3.** The experimental setup: (**a**) orientation of coordinate system and location of three points adopted for further analysis; (**b**) scheme and (**c**) photo of the measurement system based on 3D DIC [2].

### 4.2. FEM Model of the Section of Arch-Shaped Steel Sheets

Calculations covered 3 numerical models with different specificity of geometry mapping. The first model (A) (Figure 4a) was devoid of characteristic web corrugations and waviness, the second model (B) (Figure 4b) had corrugated surfaces mapped but with no waves, and the third one (C) (Figure 4c) had all the web corrugations and waves characteristic of such a profile. The models (Figure 4a–c) were developed in the ANSYS graphic module making use of the data from real element measurements. Furthermore, when describing the behaviour of the particular models, the aforementioned A, B and C signs will be used. The elastic-plastic steel model was used for the calculations, developed on the basis of the tests of steel samples [3]. The support and load conditions were accepted according to the assumptions included in the article [3]. Calculations and tests were performed in the form of axial compression of the sample.

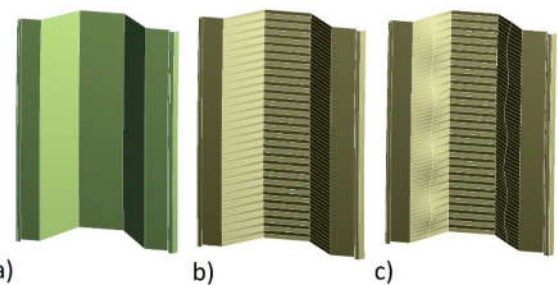

**Figure 4.** The geometry of numerical models: (**a**) planar model (A), (**b**) corrugation model (B) and (**c**) corrugation and wavy model (C).

### 4.3. Utilization of 3D DIC Measurements in the Process of Validation of FEM Model of the Section

The results of 3D DIC measurement were utilized in order to determine whether the simplification of the geometry of the model is allowed. At first, in order to perform quantitative analysis, the comparison between the displacements extracted from selected points of the structure from three models and experimental data has been performed. Points are distributed over the entire surface of the sample, in places corresponding to maximum displacements, according to FEM models. Locations of the points allowed for the assessment of representative movements for the entire sample. Exemplary data are presented in Figure 5, point 2 concerns the maximum V displacements of the sample. The best correlation between experimental and numerical results have been obtained with the FEM model comprising the most detailed geometry (C model). The results obtained from the two remaining models (of simplified geometry) differ significantly from the experimental results. Therefore, the simplifications of a FEM model have not been allowed in further analysis. Subsequently, the full-field qualitative comparison of the failure mode maps between DIC and numerical (C model) displacement maps has been carried out. Exemplary maps are presented in Figure 6. The displacement maps obtained with the numerical analysis show good agreement with the experimental data, considering its character and values. Some discrepancies that can be observed could be caused by unavoidable deviation of geometry of the sample and support conditions (e.g., inaccuracy in direction of applied force). The presented analysis validated the assumptions made in the C model, and has proven them to be useful in the analysis below.

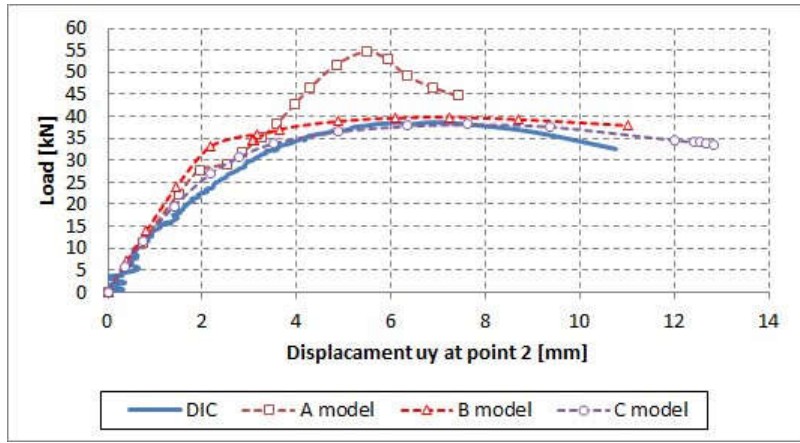

**Figure 5.** The comparison of V displacement functions obtained for three numerical models and experimental data (results for point 2) [2].

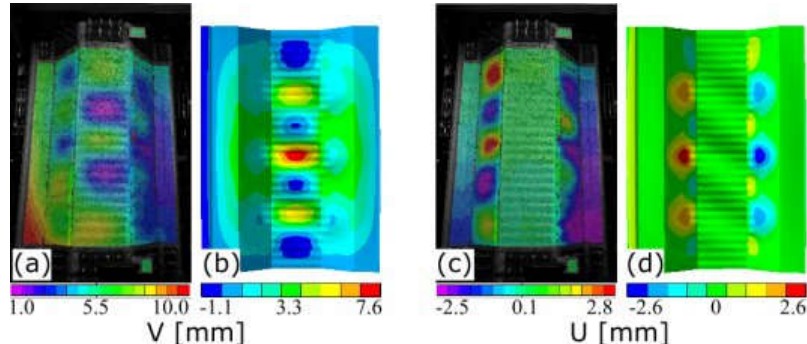

**Figure 6.** V displacement maps derived from (**a**) experiment and (**b**) the C model, and U displacement maps derived from (**c**) experiment and (**d**) the C model [2].

## 5. Investigation of Segments of Arch-Shaped Steel Sheets

The 12 m span sections of metal plate arch composed of four individual modules have been examined with the use of custom-made laboratory stands, which made it possible to apply force equivalent to a natural load caused by snow and wind [22,23]. The loading mechanism consisted of pulleys and beams which transferred point load (applied with hydraulic actuator) into 16 points (4 points for each module). The load was recorded by the actuator mounted onto the main beam, which reduced the force loss of the pulleys.

### 5.1. Multi-Camera DIC System with Overlapping Field of View

Considering the length of 12 m of arch segments, the measurements have been carried out with the use of multi-camera DIC system in which the field of view of neighboring 3D setups overlapped each other. As a result, the obtained, stitched FOV was 7 × 4 m (Figure 7a). For stitching we used the method described in the papers [13,24,25], while the general measurement procedure of the segments of arch-shaped steel sheets are presented in [13]. The multi-camera DIC system used in this measurements comprised eight 5 MPx (2448 × 2048) Pointgrey cameras equipped with 8mm focal length lenses. The cameras were connected to the control computer in order to synchronize the data acquisition procedure. The calibration procedure comprised two steps. In the first stage, 3D DIC setups have been calibrated separately with the checkerboard before the measurements. The quality of calibrations has been expressed as a reprojection error, which was smaller than 0.05 px for all 3D DIC setups. After the scaling (from pixels to mm), the accuracy of displacement measurements of each 3D DIC setup can be estimated at 0.05 mm. In the second stage, the transformation of individual coordinate systems

of separate 3D DIC systems into a common coordinate system has been determined. The fields of view of neighboring 3D DIC setups overlapped each other (Figure 7) in order to make it possible to capture the images of the same calibration target (checkerboard in this case) with two systems simultaneously. Checkerboard corners (markers) viewed by each camera have been detected and their positions in two 3D DIC coordination systems were obtained. The knowledge of position of markers in two separate coordination systems was used to obtain geometrical transformation between these two systems. The common coordination system (CS) is associated with one of the two systems. Here the coordination system of DIC setup 2 has been selected as the global CS. The transformation between CSs of setup no. 1 and 3 into CS of setup no. 2 has been determined directly. The transformation of CS of setup no. 4 was obtained indirectly—at the beginning, the transformation into CS of setup no. 3 was obtained, and then transformation from CS of setup no. 3 into CS of setup no. 2 was performed. The transformation errors obtained were below 0.5 mm. In order to correlate CSs of the measurements with numerical simulations, the data from the global coordination system (associated with setup no. 2) was transferred to the coordinate system, in which the xy plane was parallel to the ground and z axis was perpendicular to the ground (Figure 7b). AOI covers the area of corrugated web surface of two middle arches and additionally eight points on two external arches, corresponding to the locations of force extortion. Data obtained from upper (flat lip according to Figure 1) and middle (corrugated flange according to Figure 1) surfaces of the middle arches have been thresholded due to a higher correlation error (because of the loss of depth of focus).

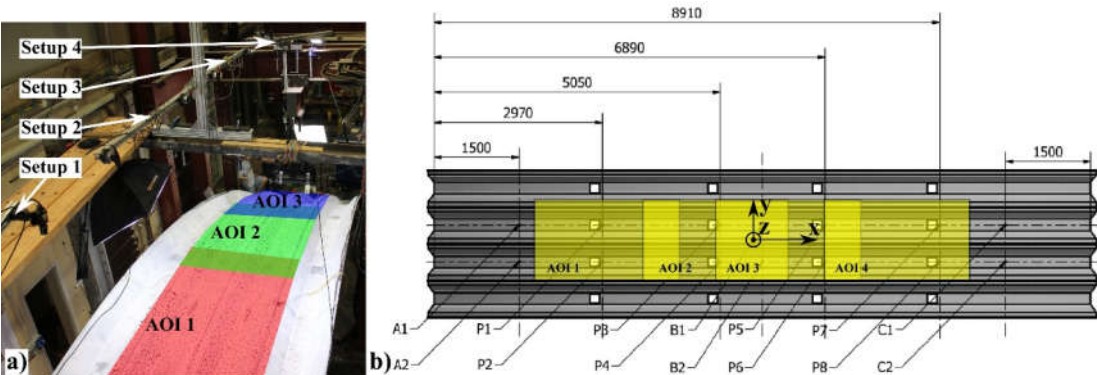

**Figure 7.** (**a**) Location of the cameras and field of views during the measurements; (**b**) scheme of the investigated segments of arch-shaped steel sheets with marked areas of interest (AOIs), characteristic points (for analysis) and orientation of coordinate system (top view) [4].

### 5.2. FEM Model of Segments of Arch-Shaped Steel Sheets

The geometry of the full-sized model has been adopted on the basis of the analysis performed in Section 4.2). The model with corrugation and waviness of the middle surfaces (model C according to the description from Section 4.2) was used for calculations. The full-sized numerical model is an image of the examined element in terms of dimensions and load mode.

In order to accurately simulate the load caused by snow cover, the kinematic extortion forced by controlled displacements of 16 points (at which load was applied) was used (Figure 7b, points A1, A2, B1, B2, C1, C2). Constraints utilized in FEM model (in particular rotational stiffness of supports) were updated through comparison with experimental data, and this process is described below.

### 5.3. Utilization of 3D DIC Measurements in Process of Validation of FEM Model of Segments of Section of Arch-Shaped Steel Sheets

Displacements of the chosen points (Figure 7b, points A1, A2, B1, B2, C1, C2) were derived on the basis of the multi-camera DIC analysis. In order to update the support conditions of the FEM model, a comparison between displacements' distribution obtained from the FEM model and

multi-camera DIC system was made. In Figure 8, an exemplary comparison has been presented that shows half of maximum load. The FEM simulation comprises two supports on both ends of the arch. Both supports are modelled by a rotational elastic spring that allows the structural member to rotate (limited by rotational stiffness), but not to translate in any direction. As the first approximation, the rotational stiffness of supports is based on the simplified theoretical calculations. Subsequently, through comparison of experimental results and theoretical calculation, the value of rotational stiffness of elastic support was determined. The results of the comparison of experimental and numerical displacements after updating the numerical one at selected points is presented on a diagram of displacements towards 3 directions (U, V, W) in the function of force increment (Figure 9). The results of the comparison of DIC tests and FEM numerical analyses are presented as balance paths with reference to proper reference points. The directions of the displacements are referred to the coordinate system as in Figure 7b. The displacements are marked as follows: U(X), V(Y) and W(Z). The horizontal displacements U (X)—Figure 9a, i.e., in the arch plane are compatible in the reference points U (3, 4), U (5, 6) and U (7, 8), whereas balance path of the reference points U (1, 2) obtained in the tests diverge from those determined in calculations. The situation is similar in vertical displacements W (Z)—Figure 9c. This means that the test element was deformed asymmetrically. The diagram comparing horizontal displacements of the reference points in the arch plane of V(Y)—Figure 9b proves that. The test model tilts erratically in the range up to 3 mm whereas the calculations show slight deviation of the reference points within 0.5 mm. The asymmetrical deformation of the test element is probably caused by imprecise assembly and slight deviations in the load symmetry. The computational model does not include random events related to the assembly or load mode. The only disorders of the geometry of the computational model are related to the introduction of geometrical imperfections, which only slightly change the symmetry of the displacements of the reference points.

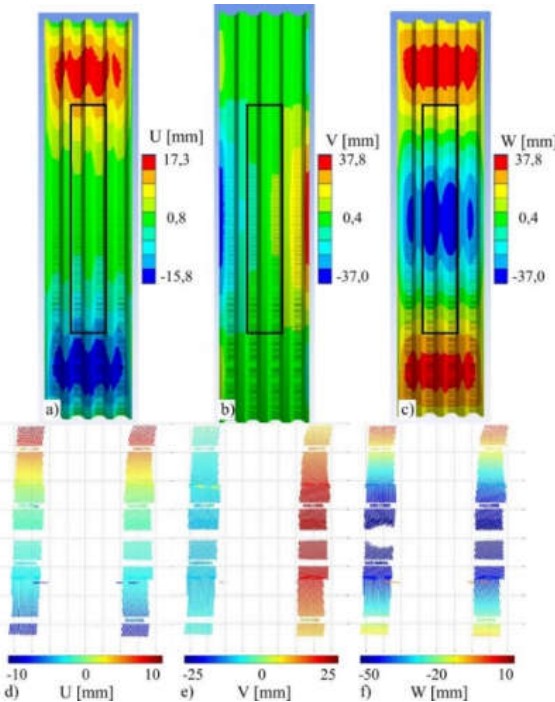

**Figure 8.** The comparison of displacements distribution: top view of displacements obtained from the finite element model (**a–c**), of the entire specimen, and experimental results (**d–f**) for the AOI covering the 7 m long area of lower surface (corrugated web according to Figure 1) of two middle arches. Rectangular areas marked with lines correspond to the areas on the experimental specimen that has the two middle arches that are being measured [4].

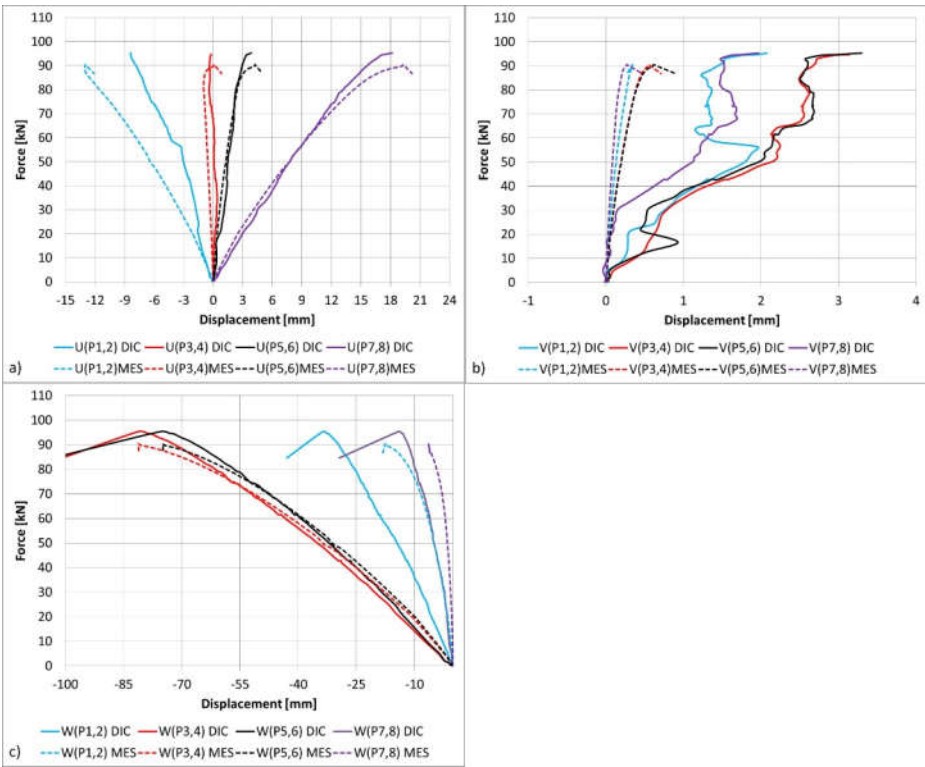

**Figure 9.** The comparison of load (**a**) U, (**b**) V, (**c**) W displacement functions obtained for numerical models and experimental data (results for referential points).

## 6. In Situ Investigation of Full-Scale Construction of Arch-Shaped Steel Sheets

Finally the full-scale construction made of arch-shaped self-supporting metal plates (Figure 1a) has been examined. The dimensions of the hall were ([span × length]/[height]) 18 × 18 m/7 m. Such a construction can be exposed to environmental loads caused by the presence of snow or wind [22,23] and changes of temperature.

### 6.1. Multi-Camera DIC System with Distributed Field of View

In order to cover the localization of points corresponding to the numerically predicted maximum displacements (caused by environmental loads), the measurements system consisted of three 3D DIC setups (six cameras in total). Each FOV of 3D DIC setup covered the area of 2.5 × 1.5 m and distance between neighboring AOIs was approximately 4 m. In order to perform the measurements in distributed FOV, the dedicated multi-camera DIC system was developed. This system and is described in detail in [16]. The multi-camera DIC system used in these measurements comprised six 5 MPx (2448 × 2048) Pointgrey cameras equipped with 8 mm focal length lenses. The cameras were connected to the control computer in order to synchronize the data acquisition procedure. The calibration procedure comprised two steps. In the first step, each 3D DIC system was calibrated with utilization of a standard chessboard calibration target. In the second calibration step, the transformation of individual coordinate systems of separate 3D DIC systems into a common coordinate system has been determined, with the additional support of a laser tracker [26]. Geometrical transformations were determined by using multiple positions of a fiducial marker, which was sequentially placed in the FOV of each 3D DIC setups. The 3D positions of the marker in local coordinate systems were determined with the use of three-dimensional computer vision algorithms [18]. Simultaneously, the positions of the marker in the global coordinate system were determined by a laser tracker. A global coordination system was related to the foundations of the building. Transformation parameters (rotation: Rx; Ry; Rz and translation: Tx; Ty; Tz) between local and global coordinate systems were obtained by using the singular value

decomposition (SVD) method [27]. The obtained accuracy of displacement measurements in each 3D DIC setup was below 0.3 mm (detailed description of calculation of displacement error is presented in [16]).

A multi-camera DIC system was placed on the 6 m high and 10 m wide buildings' temporary scaffolding (Figure 10), and the measurements were carried out for a few months (from March to October 2016). Therefore, the temperature influence on displacements of measurement setup was noticeable. In order to neutralize this influence, the displacements of the 3D DIC systems as a function of temperature, caused by the thermal expansions of aluminum scaffolding were measured by means of a laser tracker. The correction obtained was included in the results of displacement measurements. The temperature inside the hall was measured using a weather station.

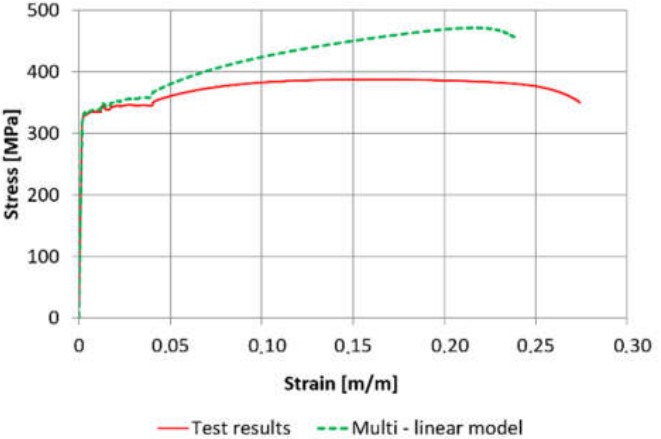

**Figure 10.** Stress–strain curves.

### 6.2. FEM Model of Full-Scale Construction

A model being the representation of the arch in the central part of the hall was used for calculations. The model consists of a single ABM 240 profile with the geometry adopted on the basis of the previous analyses. Material constants (modulus of elasticity, Poisson's modulus, shear modulus and coefficient of linear thermal expansion) have been adopted for structural analysis in accordance with EN 1993-1-1 standard [28].

The computation adopted an elastic-plastic multi-linear material model, determined according to the tests. The strength characteristics of steel were identified through a series of laboratory tests on 10 samples of steel sheet with a nominal thickness of 1.40 mm. The obtained mean yield strength amounted to $f_y$ = 340 MPa, and ultimate strength was $f_u$ = 390 MPa. A typical course of the stress–strain relationship in a single test, and elastic-plastic multi–linear material model is presented in Figure 10.

The test data were implemented via transformation functions in the $\sigma_{true}$–$\varepsilon_{ln}$ system (elastic-plastic multi-linear material model) [3] according to the Equations (1), (2) presented below:

$$\varepsilon_{ln} = ln\left(1 + \varepsilon_{eng}\right) \tag{1}$$

$$\sigma_{true} = \sigma_{eng}\left(1 + \varepsilon_{eng}\right) \tag{2}$$

where:

$\varepsilon_{ln}$—relative logarithmic strain,
$\sigma_{true}$—true stress,
$\sigma_{eng}$—engineering stress (test result),
$\varepsilon_{eng}$—engineering strain (test result).

The free ends of the model are propped up in joints (a possibility to rotate towards X axis). The remaining degrees of freedom are blocked. On the side edges, boundary conditions are assumed that map the cooperation of the adjacent profiles (remote point). In the remote point system, displacement towards Z and Y are released, the remaining degrees of freedom are blocked. Due to lack of snow during the measurement period in winter, and the negligible influence of the pressure of wind on construction, only the thermal load was considered.

The purpose of the test was to determine the displacements and stress of the characteristic points located on the surface of the test object exposed to the action of thermal loads. For numerical calculations, the external and internal temperature of arch structure from three areas (corresponding to measured AOIs) was adopted. The temperature was measured precisely with the utilization of thermoelements.

*6.3. Utilization of 3D DIC Measurements for Validation of FEM Model of Full-Scale Construction of Arch-Shaped Steel Sheets*

Displacement measurements were taken in AOI1, AOI2 and AOI3 areas and 3 reference points were selected for each area (Figure 11) to validate the numerical model, 9 points in total. The tests lasted for several months, however validation of the numerical model was limited to a much shorter period. Below (Figures 12 and 13) we present exemplary results of measurements, which were performed in 12 h and 25 min. during a cold night and sunny day in April (largest external as well as internal temperature gradient occurred). The obtained results allowed to validate the numerical model, that can be used to calculate load capacity and stability of the coverings of the thin-walled arch-shaped sheet metals.

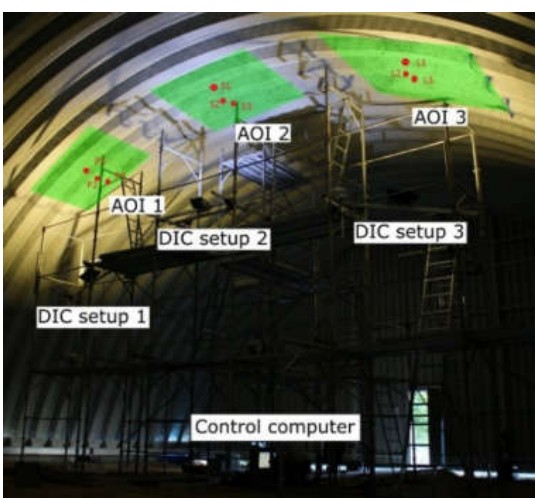

**Figure 11.** Measurement set-up.

Figure 13 presents the results of the displacements measurement and those of numerical calculations for the chosen reference points from the analysed measurement areas. The results were juxtaposed in relation to the change of temperature depending on the time of measurement. Displacements measured and determined on the basis of calculations are compatible in terms of increment directions and have similar values. Additionally, Figure 14 shows the analysis of the discrepancies of the calculation results and the test results depending on the value of temperature gradient. A certain regularity may be noticed in the distribution of discrepancies of the results. At high temperature gradients (over 18 °C), the discrepancies of neither test nor calculations results exceed 10%, at lower temperature gradients—the discrepancies of the tests results are much higher. This is probably due to the fact that numerical model did not take into account the boundary conditions related to the cooperation of the adjacent profiles, i.e., the friction between the cooperating profiles was not included in the calculations. Assuming that the friction at the jointing of the adjacent profiles has a

constant value, its influence is much more visible at small displacements than at large displacements related to the effect of the higher temperature gradient.

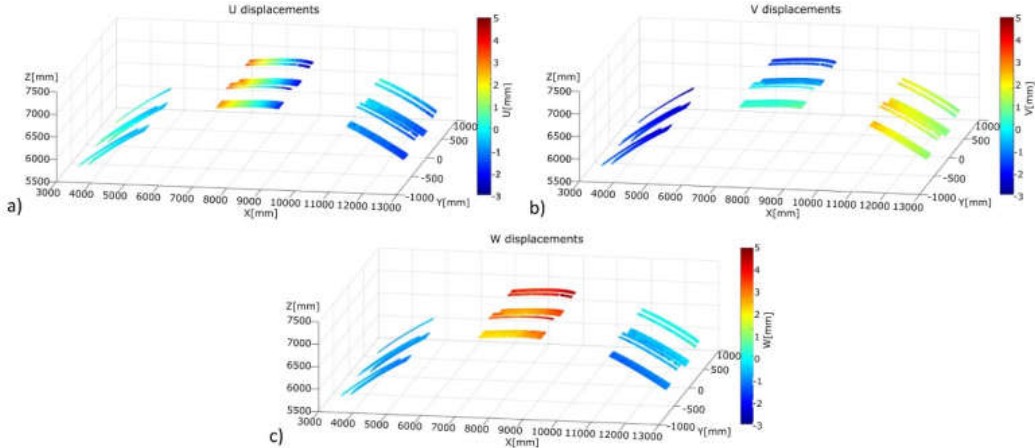

**Figure 12.** Exemplary results of the measurements stitched in the global coordinate system: (**a**) U displacement map, (**b**) V displacement map, (**c**) W displacement map [16].

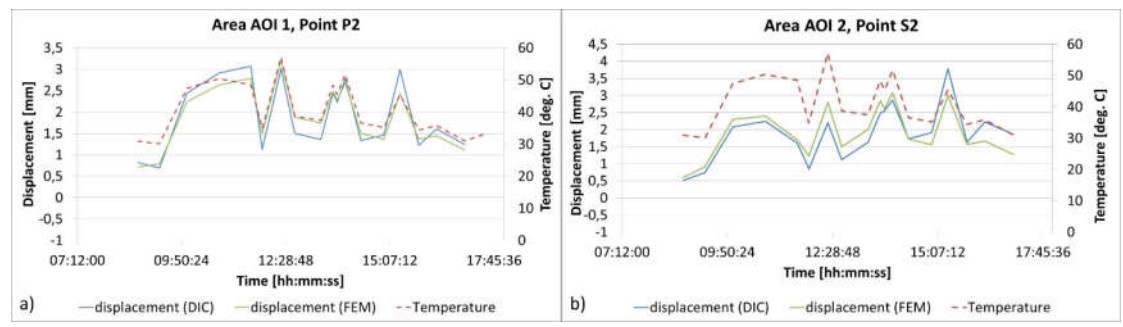

**Figure 13.** The comparison of displacement functions obtained for numerical models and experimental data, (**a**) point P2 (AOI 1), (**b**) point P2 (AOI 2).

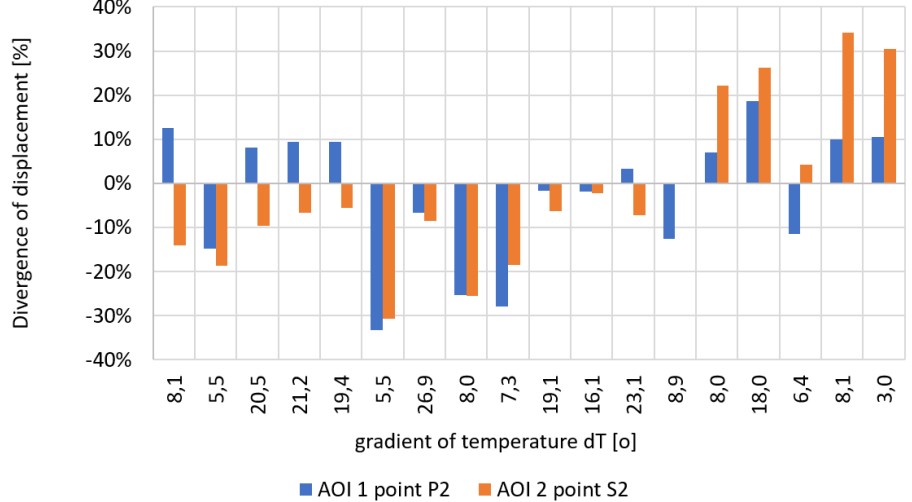

**Figure 14.** The analysis of the divergence of the measurement and calculations results.

## 7. Conclusions

So far, the DIC method applied in the construction industry has been considered a prototypical solution, more likely intended for testing/monitoring the elements of buildings, not for proper

measurements used in a certification process in accordance with accepted standards. The methodology presented in this paper has shown the possibility and advantages of replacing conventional measuring methods based at point extensometers and applied at different stages of the analysis and testing of complicated building structures by applying the 3D DIC method. Its potential lies not only in the capability to measure in full field of view, but mostly in the possibility to digitally link the measurement with the numerical calculations, thus creating efficient hybrid experimental-numerical system with huge information resources useful, for instance, in the process of design optimization, FEM validation or diagnostics of complicated building structures. It should be pointed out that the presented methodology requires a complex measurement setup and is labor-intensive, therefore it should be utilized in the case of investigation of truly complex structures, in which stress concentrations can occurr in unexpected locations.

The procedure presented in Figure 2 and explained in detail in Sections 4–6 concerns the way to implement the particular stages when analysing self-supporting arch-shaped structures from profiled steel sheets with the use of numerical calculation methods supported by physical experiments. Each stage is used to determine and validate the optimal procedure to be applied in a FEM which, in the opinion of the authors, adequately indicates the solution to the most important problems related to the design of arch-shaped structures. The multistage research verification process enables the development of a reliable numerical model that is very useful and allows for the analysis of arch-shaped profiled steel sheets at diversified geometry and any load conditions.

The methodology presented in the paper may be used to determine the strength and functional properties of various varieties of the K-span system, which are required during the process of implementing the product for use in the construction industry (Polish and European technical assessments). The material presented in the paper may also be used in the future to develop an annex to the national standards in question regarding the design of thin-walled elements.

**Author Contributions:** Conceptualization, M.K.; methodology, K.M., A.P. and M.M.; investigation K.M., M.M. and P.W.; formal analysis, K.M., M.M. and A.P., validation, K.M. and A.P.; supervision, M.K.; funding acquisition, M.K. and A.P.; visualization, K.M. and A.P.; writing—original draft, K.M. and A.P.; writing—review and editing K.M., M.K. and A.P.

**Funding:** The authors gratefully acknowledge financial support from the project OPT4-BLACH (Grant No. PBS1/A2/9/2012) financed by the National Center for Research and Development and the statutory funds of the Faculty of Mechatronics, Warsaw University of Technology.

**Conflicts of Interest:** The authors declare no conflict of interest.

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
