# Peer review of "Application of 3D Digital Image Correlation for Development and Validation of FEM Model of Self-Supporting Arch Structures"

_applsci, doi:10.3390/app9071305_

Round 1

Reviewer 1 Report

Comments

The paper proposed the hybrid experimental-numerical methodology of analysis and optimization of arch structures, which could be used especially in case of unexpected local loss of stability, which is caused by the uncommon geometry and material profile of the specimen. With the aids of Digital Image Correlation, the authors developed and validated FEM model. The methods were tested mainly with three different sizes of objects. Accuracies, sizes of fields of view, forms of outcome data and environment conditions are considered in the experiments.

General comments:

1. Although your manuscript looks well prepared, it still needs careful editing by someone with expertise in technical English editing paying particular attention to sentence structure so that the goals and results of the study are clear to the readers:

P1 L32-33, “Direct adaptation of this technology for civil purposes, combined with extending the lifetime and changing of environmental conditions, have caused…” This sentence is unclear and it’s better to rewrite it.

P1 L32, “adaptation” should be “adaptations”.

2. Typing errors: There seems to me extra space in the text of the pdf version. E.g. P3 L81 between “by” and “the”, and also Figure 2 between “of” and “numerical”.

3. The references are inadequate. Some publications with other full-field measurement applied with mechanics analysis, such as terrestrial laser scanning should be mentioned. For example, two publications are given as follows:

i) Yang et al, The Benefit of 3D Laser Scanning Technology in the Generation and Calibration of FEM Models for Health Assessment of Concrete Structures, Sensors. (related to validation)

ii) Xu et al, TLS-Based Composite Structure Deformation Analysis Validated with Laser Tracker, Composite Structures. (related to accuracy issues, sizes of fields of view, forms of outcome data, environment conditions)

The authors should review literatures of different full-field measurement methods and mention the advantages of DIC.

Other detailed comments are listed as following:

1. The author proposed a few conceptions in Figure 2, but it is not explained directly which affects the understanding of this flowchart in Figure 2. What is the meaning of “geometric model of surface shape”? Is it an arbitrary local shape or a shape of a standard part?

2. In Figure 2, why do you extract geometric model from the first step and rotational stiffness from the second step? Why do you arrange like this? What is the reason for extract kinematic extortion and use it for the second step other than the third step?

Author Response

Authors of the manuscript would like to thank the reviewer for the review.

Point 1: Although your manuscript looks well prepared, it still needs careful editing by someone with expertise in technical English editing paying particular attention to sentence structure so that the goals and results of the study are clear to the readers:

P1 L32-33, “Direct adaptation of this technology for civil purposes, combined with extending the lifetime and changing of environmental conditions, have caused…” This sentence is unclear and it’s better to rewrite it.

P1 L32, “adaptation” should be “adaptations”.

Response 1: Thank you for spotting it. We had corrected it.

Point 2: Typing errors: There seems to me extra space in the text of the pdf version. E.g. P3 L81 between “by” and “the”, and also Figure 2 between “of” and “numerical”.

Response 2: We corrected it, additionally we found and corrected some more typing errors.

Point 3: The references are inadequate. Some publications with other full-field measurement applied with mechanics analysis, such as terrestrial laser scanning should be mentioned. For example, two publications are given as follows:

i) Yang et al, The Benefit of 3D Laser Scanning Technology in the Generation and Calibration of FEM Models for Health Assessment of Concrete Structures, Sensors. (related to validation)

ii) Xu et al, TLS-Based Composite Structure Deformation Analysis Validated with Laser Tracker, Composite Structures. (related to accuracy issues, sizes of fields of view, forms of outcome data, environment conditions)

The authors should review literatures of different full-field measurement methods and mention the advantages of DIC.

Response 3: Thank you very much for this comment. The mentioned articles are very interesting and we refer to these works in the updated manuscript, by presenting brief comparison between terrestrial laser scanning and 3D Digital Image Correlation techniques in introduction (lines 45-57).

Referring to other full-field measurement methods: the article is focused on investigations of large engineering structures, therefore we had not presented a comparison of 3D DIC to the full-field measurement methods which are usually used in small fields of view (e.g. grating (moire) interferometry, digital holographic interferometry, digital speckle pattern interferometry or shearography), although one of the authors (M. Kujawinska) is applying extensively the a.m. coherent light methods for experimental mechanics and material engineering.

Point 4: The author proposed a few conceptions in Figure 2, but it is not explained directly which affects the understanding of this flowchart in Figure 2. What is the meaning of “geometric model of surface shape”? Is it an arbitrary local shape or a shape of a standard part?

Point 5: In Figure 2, why do you extract geometric model from the first step and rotational stiffness from the second step? Why do you arrange like this? What is the reason for extract kinematic extortion and use it for the second step other than the third step?

Response 4&5: Thank you very much for this comment. In section 2.2. “Three step development and validation of a numerical model” an overview of methodology is presented, the detailed description is described in the following sections (as mentioned in the manuscript). Our intention is to introduce to a reader an overview of the method in section 2 (flow-chart in figure 2 provides an overview of  the methodology), however to it is explained  step by step in the following sections. Furthermore we agree that section 2.2. was not clearly written so we had extended it and clarified the role of the sequential steps.

Reviewer 2 Report

My comments are attached. 

Author Response

Authors of the manuscript would like to thank the reviewer for the review.

Point 1: In the literature survey, the references on line 34 [1-5], line 45 [1,9-14] does not provide sufficient information about the previous works. Please use two to three sentences to explain the key findings of each cited reference and how they contribute to this study.

Response 1: Thank you for this comment. We provide information about the previous works in the updated manuscript, (lines 64-75).

Point 2: The displacement U, V, W should be clarified with corresponding coordinate X, Y, Z. Please make a clarification on figure 3a.

Response 2: Thank you for spotting it. We make a clarification in figure 3a.

Point 3: This manuscript has a long discussion with the investigation on each scale level. For the convenience of readers, please consider using a table to summarize the key finding of investigation on each scale level. For example, the maximum prediction errors from FEM upon validation to DIC measurements; the difficulties and issues for investigation on each level due to size, geometry; possible improvements.

Response 3:  Thank you for this comment. We have been also considering the use of table to summarize the features of each step of validation. However, due to many key information considering the validation procedure, such table would be also unclear for the reader. Therefore, we decide to prepare the diagram (Figure. 2), which presents key relationships in the presented methodology in intelligible way. We also modified the Section 2.2 to indicate at the beginning the goals and focus of the sequential steps of the procedure.

Point 4: Please discuss the limitation of the presented numerical-experimental method.

Response 4: Thank you for this comment. In section Conclusion the brief information about the purpose of use of the method has been added, considering the requirements for complex measurement setup (starting from line 390). Taking into account the features of 3D DIC method (scalability of sensitivity and dimensions of fields of view) and development of multi-camera DIC system with distributed FOV (the method is no longer limited to a single area of observation), the method can be used to any case of development and validation of FEM model which consider utilization of experimental displacements.

Point 5: Please format the conclusion section consistently as previous sections.

Response 5: Thank you for this comment. We format the conclusion section properly .

Reviewer 3 Report

Summary: In this paper, the author applied a so-called ‘hybrid experimental-numerical methodology’ to analysis the self-supporting arch structures used in the architecture. The finite element simulation is performed iteratively based on the comparison between the simulated results and the one from the full-field DIC (digital image correlation) technique. The calibrated properties as a function of space are obtained when the desired criteria are met. This type of work should fall into the category of the inverse problem, where a specific objective function is required. However, there is no single mathematics expression found anywhere in this manuscript. The novelty of this work is new if the author did it correctly. I have a number of points that I think the authors need to address before the paper is ready for publication. 

Detailed remarks (those that I consider more important are marked with an asterisk): 

1.∗ I strongly recommend the author to provide at least the constitutive equation for the material used. 

2.∗ To verify the inverse method, the author should compare it to a simple tension test with the known material properties. 

3.∗ What are the optimized parameters? 

 4. For similar work, the author could refer to Gokhale et al. (2008) for more details. 

References

Nachiket H Gokhale, Paul E Barbone, and Assad A Oberai. Solution of the nonlinear elasticity imaging inverse problem: the compressible case. Inverse Problems, 24(4):045010, 2008. 

Author Response

Authors of the manuscript would like to thank the reviewer for the review.

At the beginning of our reply we would like to point out that the main goal of the article (considering subject of the special issue: ‘Advances in Digital Image Correlation (DIC)’) was to present the advancements and special utilization of 3D DIC as the experimental method  in ‘hybrid experimental-numerical methodology’ applied for self-supporting arch structure. The detailed description of FEM models at each step of calibration will be provided in the further paper, which will be submitted to journal with focus on numerical methods applicable to engineering problems. Some aspects of numerical models developed for the sequential steps had been presented in publications [1, 2, 3, 13] (according to references in submitted article). In this paper we had focused mainly on the experimental part.  Nevertheless we have extended description of the final numerical model in section 6.

Point 1: I strongly recommend the author to provide at least the constitutive equation for the material used. 

Point 2: To verify the inverse method, the author should compare it to a simple tension test with the known material properties. 

Response 1&2: A relevant comments were added in section 6.2 of the manuscript (starting from line 316). The comments concern the material specification and comparison with laboratory tests.

Point 3: What are the optimized parameters? 

Response 3: A comment was added in Section 6.2  of the manuscript (starting from line 340). 

Point 4: For similar work, the author could refer to Gokhale et al. (2008) for more details. 

Response 4: The indicated reference is a valuable information, however, it does not apply to the subject of the article. In our future works, where applicable, we will refer to this interesting paper.

Round 2

Reviewer 3 Report

The author's response to my previous comments in the first round does not fully satisfy my expectation, and some response even deepens my concerns about the correctness of the proposed work. The following are some key questions that need to be considered:

1)  Section 6.2, where the author added the elastoplastic multi-linear material model for the steel. What is Young's modulus?

2) Since the author considered the multi-physics coupling in their work, where the thermally induced deformation in the structure is important, where is the balance of energy?  As for the thermal property, what is the linear thermal expansion coefficient for the steel and its thermal conductivity? 

Author Response

Authors of the manuscript would like to thank the reviewer for the review.

All the analysed research elements are made of constructional steel. The basic material constants of steel have small values specified in EN 1993-1-1 standard [28] and are, respectively:

·         Modulus of elasticity (Young)  E= 210GPa,

·         Poisson`s modulus ν = 0,3

·         Shear modulus G ≈ 81 GPa

·         Coefficient of linear thermal expansion α=12*10-6/K  (for T<100oC)

Since the calculations were performed with the assumption of short-term influence of the temperatures up to 100 °C, these parameters were not modified. Therefore, material constants recommended by EN 1993-1-1 standard [28] have been adopted for numerical calculations. The relevant comment was included in the manuscript (line 315).

Variable parameters of steel that could have affected the results of calculations are yield strength and ultimate strength. These parameters have been identified in laboratory tests and described in section 6.2.

The displacements of the structure resulting from the temperature influence are insignificant. Therefore, the stresses in the elements of the structure have negligible values included in the elastic deformability of steel. This means that the deformations are elastic and no structural changes in the steel are revealed. Issues related to balance of energy may influence the stability of the structure under significantly higher temperatures, e.g. during a fire. Detailed recommendations on this subject are considered, for example, in EN 1993-1-3 standard: Eurocode 3: Design of steel structures – Part 1-2: General rules – Structural fire design. The reviewer rightly draws attentions to the significance of this issue, however it goes beyond the subject area of this article. Further tests, including this type of phenomena in the examination of steel sheet arch structures, are planned in the future.